# TANK Binding Kinase 1 Promotes BACH1 Degradation through Both Phosphorylation-Dependent and -Independent Mechanisms without Relying on Heme and FBXO22

**DOI:** 10.3390/ijms25084141

**Published:** 2024-04-09

**Authors:** Liang Liu, Mitsuyo Matsumoto, Miki Watanabe-Matsui, Tadashi Nakagawa, Yuko Nagasawa, Jingyao Pang, Bert K. K. Callens, Akihiko Muto, Kyoko Ochiai, Hirotaka Takekawa, Mahabub Alam, Hironari Nishizawa, Mikako Shirouzu, Hiroki Shima, Keiko Nakayama, Kazuhiko Igarashi

**Affiliations:** 1Department of Biochemistry, Tohoku University Graduate School of Medicine, Sendai 980-8576, Japanhirotaka.takekawa@gmail.com (H.T.); mahabubalam@cvasu.ac.bd (M.A.);; 2Neuro-Oncology Branch, Center for Cancer Research, National Cancer Institute, Bethesda, MD 20892, USA; 3Center for Regulatory Epigenome and Diseases, Tohoku University Graduate School of Medicine, Sendai 980-8576, Japan; 4Department of Neurochemistry, Tohoku University Graduate School of Medicine, Sendai 980-8576, Japan; 5Division of Cell Proliferation, Tohoku University Graduate School of Medicine, Sendai 980-8576, Japan; tnakagaw@rs.socu.ac.jp (T.N.); nakayak2@med.tohoku.ac.jp (K.N.); 6Department of Clinical Pharmacology, Faculty of Pharmaceutical Sciences, Sanyo-Onoda City University, Sanyo-Onoda 756-0884, Japan; 7Faculty of Health, Medicine and Life Sciences, Maastricht University, 6229 GT Maastricht, The Netherlands; 8Department of Animal Science and Nutrition, Chattogram Veterinary and Animal Sciences University, Khulshi, Chattogram 4225, Bangladesh; 9Laboratory for Protein Functional and Structural Biology, RIKEN Center for Biosystems Dynamics Research, Yokohama 305-0074, Japan

**Keywords:** autophagy, BACH1, TBK1, FBXO22, heme, phosphorylation, ubiquitination

## Abstract

BTB and CNC homology 1 (BACH1) represses the expression of genes involved in the metabolism of iron, heme and reactive oxygen species. While BACH1 is rapidly degraded when it is bound to heme, it remains unclear how BACH1 degradation is regulated under other conditions. We found that FBXO22, a ubiquitin ligase previously reported to promote BACH1 degradation, polyubiquitinated BACH1 only in the presence of heme in a highly purified reconstitution assay. In parallel to this regulatory mechanism, TANK binding kinase 1 (TBK1), a protein kinase that activates innate immune response and regulates iron metabolism via ferritinophagy, was found to promote BACH1 degradation when overexpressed in 293T cells. While TBK1 phosphorylated BACH1 at multiple serine and threonine residues, BACH1 degradation was observed with not only the wild-type TBK1 but also catalytically impaired TBK1. The BACH1 degradation in response to catalytically impaired TBK1 was not dependent on FBXO22 but involved both autophagy-lysosome and ubiquitin-proteasome pathways judging from its suppression by using inhibitors of lysosome and proteasome. Chemical inhibition of TBK1 in hepatoma Hepa1 cells showed that TBK1 was not required for the heme-induced BACH1 degradation. Its inhibition in Namalwa B lymphoma cells increased endogenous BACH1 protein. These results suggest that TBK1 promotes BACH1 degradation in parallel to the FBXO22- and heme-dependent pathway, placing BACH1 as a downstream effector of TBK1 in iron metabolism or innate immune response.

## 1. Introduction

Transcription factors are crucial proteins in cells because they control gene expression, thereby influencing cell function and organism development. The amount and dynamics of iron within cells are critical for their metabolism, survival, proliferation, and death, and are regulated not only at the post-transcriptional levels involving iron regulatory proteins (IRP1 and IRP2) but also at the transcriptional level in mammalian cells [1]. The transcription repressor BACH1 regulates the expression of genes involved in the metabolism of iron, heme, and reactive oxygen species (ROS), and regulates the execution of ferroptosis, an iron-dependent cell death [2,3,4,5]. For example, BACH1 represses the expression of ferritin and the iron exporter ferroportin to increase intracellular free iron [2,6,7]. BACH1 also represses the expression of the heme oxygenase-1 gene to catabolize heme and recycle iron. Importantly, heme inhibits the repressor activity of BACH1 by directly binding to BACH1; heme inhibits its DNA binding activity, induces its nuclear export, and promotes polyubiquitination and proteasome-mediated degradation [8,9,10]. Heme thus induces the expression of BACH1 target genes. However, the expression of these BACH1 target genes is also altered in response to signals other than heme. For example, the ferritin gene is induced in response to inflammation and infection [11,12]. Therefore, there may be additional mechanisms, other than heme, in the regulation of BACH1 activity.

BACH1 and its related factor BACH2 repress the expression of innate immune genes in hematopoietic stem and progenitor cells, thereby promoting their differentiation into B and T lymphocytes [13,14] or to red blood cells [15]. BACH1 promotes the malignancy of cancer cells, including metastasis, in breast cancer [16,17], pancreatic ductal adenocarcinoma [18], and lung cancer [19,20]. These multiple functions of BACH1 in hematopoietic and cancer cells point to the importance of BACH1 regulation and suggest the presence of mechanisms for BACH1 regulation other than heme.

BACH1 is polyubiquitinated by the E3 ligase adaptor proteins F-box protein 22 (FBXO22) [19], F-Box and leucine rich repeat protein 17 (FBXL17) [21] and heme-oxidized IRP2 ubiquitin ligase 1 (HOIL-1) [10], and is degraded via the ubiquitin-proteasome pathway. While HOIL-1 was shown to polyubiquitinate BACH1 in response to heme in a purified reconstituted system [10], it remains unclear whether the recognition of BACH1 by FBXO22 or FBXL17 is directly regulated by heme. Still unknown is whether BACH1 degradation is regulated by signals other than heme. Whether BACH1 is degraded solely by the proteasome system also needs to be explored.

TANK binding kinase 1 (TBK1) is a serine/threonine kinase with an important role in multiple signaling pathways [22,23]. The majority of research on TBK1 has focused on its role in innate immunity. cGAS-STING pathway activates TBK1, which then activates the transcription factors IRF3 and NF-κB, resulting in the production of antiviral and proinflammatory cytokines, including type I interferons [24,25,26]. In addition, the role of TBK1 has expanded into cancers, autophagy and ubiquitination [27,28,29,30,31,32,33,34,35]. TBK1 has been reported to promote autophagy-mediated degradation of ferritin which is important for the regulation of iron homeostasis [36]. One study showed that TBK1 ubiquitinates multiple picornavirus VP3 proteins as an E3 ubiquitin ligase [37]. Given that BACH1 regulates the expression of ferritin genes [2,6] and that its multiple serine and threonine residues are phosphorylated [38], we hypothesized that TBK1 may also regulate BACH1 protein. This idea was further prompted by the recent finding that BACH1 mRNA expression is dependent in part on TBK1 [39]. We examined here whether TBK1 is involved in the regulation of BACH1 protein including its phosphorylation and degradation. TBK1 promoted the degradation of BACH1 in both phosphorylation-dependent and -independent manners. Importantly, the effects of TBK1 upon BACH1 was found independent of its heme-regulated, FBXO22-mediated polyubiquitination. 

## 2. Results

### 2.1. FBXO22 Mediates Polyubiquitination of Heme-Bound BACH1

While it has been reported that FBXO22 promotes BACH1 degradation in cells when heme is increased [19], it remains unclear if heme collaborates with FBXO22 to promote Bach1 degradation. We confirmed the interaction of FBXO22 with BACH1 by immunoprecipitation and mass spectrometry analysis (Figure 1A; Appendix A). Co-expression of FBXO22 along with BACH1 decreased the amount of BACH1 protein in HEK293T cells (Figure 1B). When FBXO22 and His-tagged ubiquitin were co-expressed, followed by treatment with MG132, a proteasome inhibitor, there was a marked increase in the polyubiquitination of BACH1. This finding suggested that FBXO22 serves as an important facilitator for the addition of multiple ubiquitin molecules to BACH1, targeting it for proteasomal degradation (Figure 1C). The endogenous regulation of BACH1 protein amount by FBXO22 was further examined by using human B cell line Namalwa cells derived from B cell lymphoma. Knockdown of FBXO22 prolonged the half-life of endogenous BACH1 in human B cells (Appendix A).

To examine whether BACH1 is directly ubiquitinated by FBXO22, we established an in vitro ubiquitination assay using purified proteins. Importantly, the in vitro reconstituted ubiquitination assay demonstrated that FBXO22 promoted polyubiquitination of BACH1 only when heme was added, with clear dose-dependency. As the concentration of heme increased to 4 μM, the ubiquitination became progressively more pronounced (Figure 1D,E). These results with the purified reconstitution system established for the first time that FBXO22 mediated polyubiquitination of BACH1 and heme enhances polyubiquitination. 

### 2.2. TBK1 and FBXO22 Constitute Separate Pathways for BACH1 Turnover

We have recently found that TBK1 increases the BACH1 mRNA amount in cells [39] and that TBK1 phosphorylates BACH2 (M. Matsui-Watanabe et al., submitted). Because BACH1 and BACH2 are highly similar in structure [40], we examined whether TBK1 also regulated BACH1. We examined alterations in the phosphorylation status of BACH1 by overexpressing TBK1 in HEK293T cells. When we prepared cell lysates 48 h after transfection, both TBK1 and TBK1 S172A, a mutant TBK1 defective in autoactivation and hence phosphorylation activity [41], decreased the amount of BACH1 protein (Figure 2A). The co-expression of TBK1 also caused a shift of the BACH1 band upon immunoblotting. Since this shift of the BACH1 band was abolished by treating the cell lysates with calf intestine alkaline phosphatase (CIAP), TBK1 phosphorylated BACH1 and regulated its turnover. Surprisingly, TBK1 S172A was found to decrease BACH1 protein without its phosphorylation. 

We combined TBK1 S172A expression and FBXO22 knockdown to assess their epistatic relationship (Figure 2B). Upon knockdown of FBXO22, the amount of BACH1 protein was increased. TBK1 S172A still decreased the amount of BACH1 protein even when FBXO22 was greatly reduced. This may reflect a negative feedback regulation to maintain the amount of BACH1 protein. Next, the relationship between TBK1-mediated phosphorylation and FBXO22-mediated polyubiquitination was examined using the purified reconstitution system. Prior phosphorylation of recombinant BACH1 by TBK1 did not increase its polyubiquitination by FBXO22 (Figure 2C). Taken together, these results suggested that heme-responsive, FBXO22-dependent degradation of BACH1 was separable from its TBK1-mediated degradation; TBK1 and FBXO22 constitute separate pathways in the degradation of BACH1 protein. 

### 2.3. TBK1 S172A Promotes BACH1 Degradation through Autophagy-Lysosome and Ubiquitin-Proteasome Pathways

We investigated whether the autophagy-lysosome or ubiquitin-proteasome pathway is involved in the effect of TBK1 on BACH1. Cells were transfected with plasmids expressing TBK1 S172A and BACH1, and the autophagy-lysosome inhibitor chloroquine (CQ) or the proteasomal inhibitor MG132 was administered for the last 12 h before cell lysate preparation. When cell lysates were prepared 24 h after the transfection, CQ reversed the effect of TBK1 S172A on BACH1 (Figure 3A). When lysates were prepared 48 h after transfection, MG132 also reversed the effect of TBK1 S172A (Figure 3B). The effects of these inhibitors were found dependent on the time points when cell lysates were prepared. The effect of CQ on the BACH1 reduction in response to TBK1 S172A was clearly observed 24 h after transfection, but not thereafter (Figure 3C). In contrast to CQ, while MG132 failed to block BACH1 reduction 24 h after transfection, it efficiently prevented the reduction of BACH1 at 36 and 48 h (Figure 3D). These results indicated that TBK1 S172A promoted degradation of BACH1 by both autophagy-lysosome and ubiquitin-proteasome pathways when overexpressed in HEK293T cells. In addition, while BACH1 degradation was mainly mediated by autophagy-lysosome pathway within 24 h after transfection, its degradation was switched to ubiquitin-proteasome pathway thereafter. In other words, TBK1 S172A was able to use two pathways after transfection to promote the degradation of BACH1 in this model. 

### 2.4. TBK1 Promotes BACH1 Phosphorylation

Co-expression of wild-type TBK1 caused a more obvious shift of the BACH1 band upon immunoblotting when cell lysates were prepared 24 h after transfection of the expression plasmids (Figure 4A) than when they were prepared at 48 h after transfection (see Figure 2A). However, the amount of the BACH1 protein was not altered in response to TBK1 at 24 h (Figure 4A). As expected, TBK1 S172A did not show a shift of the BACH1 band but reduced the amount of BACH1 protein (Figure 4A). The effects of TBK1 and TBK1 S172A on BACH1 were not observed with unrelated protein kinase SRC; it did not cause a shift or a reduction of the BACH1 band (Figure 4A). 

Using this experimental condition, we determined phosphorylation sites of BACH1 by mass spectrometry analysis of BACH1 co-expressed with TBK1 or TBK1 S172A in HEK293T cells. Twenty-seven serine/threonine residues were phosphorylated when BACH1 was expressed alone, whereas the additional 23 sites were phosphorylated by TBK1 but not TBK1 S172A (Figure 4B). When analyzing the amino acids surrounding each of these 23 sites, we found that the TBK1-induced phosphorylated sites of BACH1 were followed by residues such as serine, aspartic acid, lysine, glutamine and hydrophobic residues, including phenylalanine, isoleucine, leucine (Figure 4C). The pattern of these sequences is in line with a previous report on the phosphorylated target sites of TBK1 [42]. These results suggested that TBK1 phosphorylated BACH1. These results also confirmed that TBK1 S172A was impaired in substrate phosphorylation.

### 2.5. TBK1-Mediated Phosphorylation Diverts BACH1 from Autophagy and Proteasome

Next, we examined whether BACH1 degradation induced by the wild-type TBK1 involved autophagy-lysosome and ubiquitin-proteasome pathways. By using the same experimental conditions as above, CQ or MG132 did not efficiently increase the amount of BACH1 protein (Appendix A). It should be noted that the band of phosphorylated BACH1 was decreased in the presence of these inhibitors (Appendix A), which is discussed later. Even when both of the inhibitors were combined, there was only a small additive effect on BACH1 (Appendix A). These results suggested that BACH1 which was phosphorylated by TBK1 was degraded independently of autophagy-lysosome or ubiquitin-proteasome pathway. We concluded that TBK1 promoted BACH1 degradation by two distinct mechanisms which were kinase activity-independent and -dependent. 

### 2.6. Endogenous Regulation of BACH1 by TBK1

To investigate the biological significance of our above findings, we inhibited TBK1 with two inhibitors in the presence or absence of exogenous heme and examined whether BACH1 protein level would be altered. As shown in Figure 5A, the treatment of Hepa1 hepatoma cells with TBK1 inhibitors did not affect the amount of BACH1 mRNA. In contrast, BACH1 protein (upper band) was increased in response to the drugs (Figure 5B). TBK1 inhibitors failed to increase BACH1 when heme was present. This is consistent with the above results showing the independence of FBXO22 on TBK1 (see Figure 2B). When the human B cell line derived from B cell lymphoma was treated with the TBK1 inhibitors, BACH1 protein amount was increased (Figure 5C), suggesting the universality of BACH1 protein regulation by TBK1. These results together support the endogenous regulation of BACH1 protein amount by TBK1 in cells.

## 3. Discussion

In this study, we found that TBK1 phosphorylates BACH1 and promotes its degradation by co-expressing these proteins in HEK293T cells. Importantly, TBK1 S172A whose phosphorylation activity is severely reduced also promoted the degradation of BACH1. Two lines of observations suggest the presence of two separable mechanisms for the TBK1-mediated regulation of BACH1 degradation. The wild-type TBK1 promoted the degradation of BACH1 even when autophagy-lysosomal and ubiquitin-proteasome pathways were inhibited, whereas TBK1 S172A-promoted BACH1 degradation was dependent on the two pathways. Therefore, TBK1 promotes degradation of BACH1 by phosphorylation-dependent and -independent mechanisms. TBK1 promoted degradation of BACH1 without depending on FBXO22 and heme. Vice versa, TBK1 was not required for the heme-promoted BACH1 degradation. Therefore, the regulation of BACH1 by its degradation is more complex than previously thought, which suggests BACH1 is an integrator of multiple signals besides heme. 

The phosphorylation-dependent degradation and phosphorylation-independent degradation of BACH1 by TBK1 appear to be mutually exclusive. Since phosphorylation-dependent degradation in response to the wild-type TBK1 was not abrogated upon inhibition of proteasome or lysosome, this pathway may involve other degradation systems, such as the calpain system [43]. It should be noted that TBK1 phosphorylates cargo receptors of selective autophagy, promoting their interactions with respective cargos, such as damaged mitochondria and lysosomes, leading to selective degradation via autophagy [35,44,45,46,47]. Such interactions of TBK1 with cargo receptors may also contribute to BACH1 degradation via autophagy. The detailed mechanism for this phosphorylation-dependent degradation pathway awaits further studies. Regarding the phosphorylation-independent effect of TBK1, we found that FBXO22 was not involved in the TBK1 S172A-promoted degradation of BACH1. The identity of the ubiquitin E3 ligase for the TBK1-promoted, phosphorylation-independent degradation of BACH1 likely involves other E3 ligase adaptors. TBK1 was reported previously to function as an E3 ubiquitin ligase of multiple picornavirus VP3 proteins [37]. However, in contrast to our results, the function of TBK1 as an E3 ligase was dependent on its kinase activity [37], leaving this possibility of BACH1 degradation less likely. Hence, further studies are required to identify ubiquitin E3 ligases that mediate phosphorylation-independent degradation of BACH1 in response to TBK1.

Our present results suggest that TBK1 promotes BACH1 degradation both when it is active as a protein kinase and when it is inactive. While both mechanisms reduce the amount of BACH1 protein, there may be differences in regulatory consequences. For example, BACH1 phosphorylation by TBK1 may also alter its interaction with co-repressors [38,48,49]. Since the cellular context and stress signals are known to significantly affect TBK1 [23,50], the influence of TBK1 on BACH1 may qualitatively be altered depending on its protein kinase activity. Depending on the TBK1 kinase activity, kinetics and/or the extent of BACH1 inactivation and degradation may vary. It will be an important issue to identify cellular conditions that utilize one or the other of the two TBK1 mechanisms for BACH1 regulation. Such a combination of the degradation of BACH1 and reduced interactions with corepressors is expected to lead to more efficient and rapid induction of BACH1 target genes. A comparison of the two pathways in terms of target gene regulation will be critical for further understanding the physiological significance of the TBK1-BACH1 pathway. Interestingly, phosphorylation of BACH1 by TBK1 was rather reduced upon inhibition of proteasome or lysosome, indicating that both proteasome and lysosome somehow contribute to BACH1 phosphorylation by TBK1. One possibility is that proteasome and lysosome both not only promote BACH1 degradation when it is not phosphorylated but also promote its phosphorylation by TBK1. Because TBK1 decreased BACH1 even when proteasome and lysosome were inhibited, there must be another protein degradation system for BACH1. NFE2L1, a transcription factor distantly related to BACH1, is degraded by both proteasome and autophagy. When proteasome is inhibited, autophagy-mediated degradation is induced [51]. In a similar light, there could be a compensation of BACH1 degradation between proteasome and autophagy.

We reported recently that TBK1 maintains the amounts of BACH1 mRNA and protein in pancreatic cancer cells [39]. Considering that BACH1 promotes cancer progression [16,17,18,19], the negative regulatory role of TBK1 on BACH1 found in the present study may be modulated in cancer cells to a positive regulatory role, promoting BACH1 protein accumulation. 

At present, the conditions under which TBK1 regulates BACH1 protein remains unclear. Our results clearly showed that TBK1 promoted BACH1 degradation under normal cell culture conditions in Hepa1 and Namalwa cells. Hence, TBK1 appears to participate in the homeostatic control of BACH1 protein amount in different types of cells. In addition, given that TBK1 is activated in response to cytoplasmic DNA to regulate innate immune response [24,25], the degradation of BACH1 may be increased as a part of the innate immune response. This possibility, including the physiological role of the TBK1-BACH1 axis in innate immune response, requires further study. Another interesting possibility is iron metabolism, since TBK1 is known to promote ferritinophagy [36]. Under such conditions, BACH1 may be degraded; ferritin gene expression may be increased to compensate for the reduction of ferritin proteins.

## 4. Materials and Methods

### 4.1. Reagents

MG132 was purchased from Calbiochem (474790, San Diego, CA, USA). Autophagy-lysosome inhibitor chloroquine (CQ) was purchased from Sigma-Aldrich (C6628, St. Louis, MO, USA). Ammonium bicarbonate (1 mol/L, 012-21745), methanol (67-56-1), acetonitrile (012-19851) and formic acid (067-04531) for mass spectrometry were purchased from FUJIFILM Wako Pure Chemical Industries (Osaka, Japan). BX795 (S1274, InvivoGen, San Diego, CA, USA) and MRT67307 (SML0702, Sigma-Aldrich) were dissolved in dimethyl sulfoxide (DMSO).

### 4.2. Cells and Cell Culture 

Human embryonic kidney 293T (HEK293T) cells and Hepa1 hepatoma cells were maintained in DMEM-low/high glucose supplemented with 10% heat-inactivated FBS (172012, Sigma-Aldrich, St. Louis, MO, USA), 100 U/mL penicillin and 100 μg/mL streptomycin (15140-122, penicillin-streptomycin, Thermo Fisher Scientific, Waltham, MA, USA). Namalwa cells were cultured in suspension in RPMI medium (Sigma-Aldrich) supplemented with 10% FBS, 100 U/mL penicillin and 100 μg/mL streptomycin. The cells used were limited to less than 20 passages.

### 4.3. Western Blotting

Cells were washed with PBS and harvested by centrifugation at 2300× *g* for 1 min. The cells were lysed using RIPA buffer (150 mM NaCl, 50 mM Tris-HCl, 1% NP-40, 0.5% sodium deoxycholate and 0.1% SDS) for 30 min on ice. After centrifugation at 20,400× *g* for 30 min, the supernatant was mixed with SDS sample buffer (62.5 mM Tris-HCl pH 6.8, 1% SDS, 10% glycerol, 1% 2-mercaptoethanol and 0.02% bromophenol blue) containing protease inhibitors (0469315900, cOmplete^®^ Mini EDTA-free Protease Inhibitor Cocktail Tablets, Roche, Mannheim, Germany). The samples were separated by SDS-PAGE using 7.5% or 10% acrylamide gels at 100 V for about 2 h and wet-transferred onto polyvinylidene difluoride (PVDF) membranes at 300 mA for 1.5 h at 4 °C. The blots were washed with TBS-T (25 mM Tris, 137 mM NaCl, 3 mM KCl, 0.05% Tween-20, pH 7.4) and incubated in TBS-T containing 5% skimmed milk for 1 h for blocking. The blots were incubated with the primary antibodies diluted in TBS-T containing 5% skimmed milk overnight at 4 °C. The blots were washed three times for 10 min in TBS-T and incubated with the horseradish peroxidase (HRP)-conjugated secondary antibodies diluted in TBS-T containing 5% skimmed milk for 1 h at room temperature. Luminescence detection was performed using a Clarity Western ECL substrate (1705060, Bio-Rad Laboratories, Hercules, CA, USA) in a ChemiDoc Touch imaging system (version 3.1, Bio-Rad Laboratories) after washing the membrane three times for 10 min with TBS-T. Quantification of the Western blotting results was conducted by measuring the band density on the same image using the ImageJ software (version 1.8.0, last accessed on 1 March 2024, http://imagej.nih.gov/ij/; provided in the public domain by the National Institutes of Health, Bethesda, MD, USA) [52].

### 4.4. Antibodies

The BACH1 protein was detected by Western blotting using either the mouse anti-BACH1 monoclonal antibody 9D11 (1:500, generated in-house) or rabbit anti-BACH1 antiserum A1–6 (1:1000, generated in-house) as reported previously [18,53]. Other antibodies used in the experiments were as follows: rabbit anti-ACTB (1:1000, GTX109639, GeneTeX, Inc. Irvine, CA, USA), rabbit anti-TBK1 (1:1000, D1B4, Cell Signaling, Danvers, MA, USA), mouse anti-SRC (1:1000, 130124, Santa Cruz Biotech, Santa Cruz, CA, USA), rabbit anti-FBXO22 (1:1000, 13606-1-AP, Proteintech, Rosemont, IL, USA), mouse anti-DDDDK-tag (1:1000, M185-3L, MBL Life Science, Woburn, MA, USA), rabbit anti-His-tag (1:1000, PM032, MBL Life Science, Nagoya, Japan) and rabbit anti-Ubiquitin (1:1000, MFK-003, Nippon Bio-Test Laboratories Inc., Tokyo, Japan). HRP-conjugated anti-rabbit IgG (1:2500, NA934V, GE Healthcare, Fairfield, CT, USA) and HRP-conjugated anti-mouse IgG (1:2500, NA931V, GE Healthcare, Fairfield, CT, USA) were used as secondary antibodies.

### 4.5. siRNA

Target-specific siRNAs (Stealth RNAi siRNA Duplex Oligoribonucleotides, Thermo Fisher Scientific, Waltham, MA, USA) were transfected using Lipofectamine RNAiMAX (13778100, Thermo Fisher Scientific, Waltham, MA, USA). Stealth RNAi siRNA Negative Control, Low GC (12935110, Thermo Fisher Scientific, Waltham, MA, USA) was used as the control siRNA.
siTBK1-15′-GCGAGAUGUGGUGGGUGGAAUGAAU-3′siTBK1-25′-GGGAACCUCUGAAUACCAUAGGAUU-3′siFBXO22-15′-UCGUGUGGUCCUUGUCUUUGGUUAU-3′siFBXO22-25′-GCUGUAAGGUGGGAGCCAGUAAUUA-3′

### 4.6. Overexpression

HEK-293T cells were transfected with indicated plasmids using GeneJuice (70967, Novagen, San Diego, CA, USA) and Opti-MEM (51985, Life Technologies, Inc., Carlsbad, CA, USA) according to the reagent protocol. 

### 4.7. Quantitative Real-Time PCR

RNA was isolated using RNeasy Plus Mini Kit (74136, Qiagen, Valencia, CA, USA) and 500 ng of total RNA was reverse transcribed to single-stranded cDNA using the High-Capacity cDNA Archive Kit (4368814, Applied Biosystems, Foster City, CA, USA). Quantitative PCR was performed with LightCycler Fast Start DNA Master SYBR Green I (06924204001, Roche, Basel, Switzerland) in LightCycler 96 instrument (Roche, Basel, Switzerland). We chose *ACTB* as the housekeeping gene. The primer sequences were: BACH1 (5′-GCCCGTATGCTTGTGTGATT-3′ and 5′-CGTGAGAGCGAAATTATCCG-3′), ACTB (5′-CGTTGACATCCGTAAAGACCTC-3′ and 5′-AGCCACCGATCCACACAGA-3′).

### 4.8. Immunoprecipitation

Cells were lysed with a lysis buffer composed of 50 mM HEPES, pH 8, 250 mM KCl, 0.2 mM EDTA, 0.1% Triton X-100, and protease inhibitors. After sonication, lysates were centrifuged and the supernatant was used as whole cell lysates. The lysate was incubated with anti-FLAG beads (M185-11R, MBL, Tokyo, Japan) on a rotating platform for 2 h at 4 °C. The sample was eluted with FLAG peptide (F4799, Sigma-Aldrich, Dallas, TX, USA). In affinity-purification of ubiquitin shown in Figure 5, cells were lysed with a buffer composed of 20 mM Tris-HCl, pH 8, 500 mM NaCl, 5 mM Imidazole, 8 M urea, and protease inhibitors. Proteins modified with 6×-His-Ub was purified using Ni-NTA Sepharose (142350243, Qiagen, Alameda, CA, USA) on a rotating platform for 2 h at 4 °C. The sample was eluted with 500 mM Imidazol.

### 4.9. Purification of His-FLAG-TBK1

Baculovirus was generated by transfecting purified bacmid DNA into Sf9 cells using FuGENE HD (E1910, Promega Corporation, Madison, WI, USA), and subsequently used to infect suspension cultures of Sf9 cells (2 × 10^6^ cells/mL) with a multiplicity of infection of 1. Infected Sf9 cells were incubated at 27 °C for 48 h for protein expression. The Sf9 cells were suspended in buffer A (20 mM Tris-HCl, pH 8.0, 150 mM NaCl, 10% glycerol, and 1 × protease inhibitor), sonicated at 4 °C and centrifuged at 28,980× *g* for 20 min. Soluble protein fractions were mixed with FLAG-M2-agarose beads (A2220, Sigma-Aldrich, St. Louis, MI, USA) pre-equilibrated with buffer A. The His-FLAG-TBK1 protein was eluted using buffer A containing 3 × FLAG peptide (200 µg/mL), concentrated using an Amicon Ultra-15 centrifugal filter unit (Merck, Billerica, MA, USA), and applied to a G-25 spin column (Cytiva) pre-equilibrated with buffer including 20 mM Tris-HCl pH 8.0, 150 mM NaCl, 1 × protease inhibitor, and 1 × phosphatase inhibitor (PhosSTOP, 04906837001, Roche, Basel, Switzerland).

### 4.10. Purification of MBP-BACH1

The expression plasmid of MBP-BACH1 was described previously [54]. The *E*. *coli* cells were suspended in buffer A (20 mM Tris-HCl (pH 7.5), 200 mM NaCl, 1 mM EDTA, 1 mM DTT, 1 × protease inhibitor and 10% glycerol), sonicated at 4 °C and centrifuged at 28,980× *g* for 20 min. Soluble protein fractions were mixed with an amylose resin column (New England BioLabs, Beverly, MA, USA) pre-equilibrated with buffer A. The MBP-BACH1 protein was eluted using buffer A containing 10 mM maltose, concentrated using an Amicon Ultra-15 centrifugal filter unit (Merck, Billerica, MA, USA), and applied to a PD10 spin column (BD) pre-equilibrated with buffer including 20 mM HEPES (pH7.0), 2 mM TCEP, 50 mM NaCl, 10% glycerol and protease inhibitor.

### 4.11. In Vitro Kinase Assay

Purified MBP-BACH1 (10 µM) and His-FLAG-TBK1 (0.5 µM) were used for the assay. The reactions were performed using Kinase Buffer (9802, Cell Signaling Technology, Danvers, MA, USA), 0.1 mM ATP, supplemented with 1 × protease inhibitor and 1 × phosphatase inhibitor at 30 °C for 30 min. After the phosphorylation reaction, the in vitro ubiquitination assay was carried out.

### 4.12. In Vitro Ubiquitylation Assay

HA-BACH1 was expressed and purified from HEK293T cells; HEK293T cells were plated on four 15 cm dishes at a density of 6–7 × 10^6^ cells per dish in DMEM containing 10% FBS. After 24 h incubation, cells on every dish were transfected with 15 μg of pcDNA3-HA-BACH1 using the PEI MAX reagent (24765, Polysciences, Warrington, PA, USA). At 48 h after transfection, cells were lysed with an NP-40 buffer (50 mM Tris-HCl pH 7.5, 150 mM NaCl, 0.5% Nonidet *p*-40, 10% Glycerol, and a protease inhibitor cocktail (10 μg/mL aprotinin, 10 μg/mL leupeptin, and 1 mM phenylmethylsulfonyl fluoride) and then HA-BACH1 was purified from the resulting lysates with the use of HA tagged Protein Purification Kit (3320A, MBL, Nagoya, Japan).

The ubiquitin E2 enzymes CDC34 and UBCH5C were purified from bacteria. *E. coli* (BL21) transformed with a plasmid encoding His6-tagged either CDC34 or UBCH5C were cultured at 37 °C to an optical density at 600 nm of 1.0. The temperature was then lowered to 30 °C, and the cells were exposed to 0.1 mM isopropyl-D-1-thiogalactopyranoside for 6 h, harvested, and then lysed with the use of B-PER (Thermo Scientific). After removal of debris by centrifugation, the lysates were incubated with Ni-nitrilotriacetic acid (NTA) beads (Amersham Biosciences, Amersham, UK) to precipitate His6-tagged E2s, which were then eluted with 250 mM imidazole.

SCF-FBXO22 was co-purified as follows. FLAG-tagged FBXO22 and associated proteins were immunoprecipitated from transfected HEK293T cells with an anti-FLAG antibody, and then eluted by incubation with the FLAG peptide with the use of FLAG tagged Protein Purification Kit (MBL, Tokyo, Japan).

Various combinations of E1 (100 ng; Boston Biochem), E2 (220 ng of CDC34 and 110 ng of UBCH5C), and E3 (90 ng of SCF-FBXO22) were mixed with 60 ng of recombinant HA-BACH1 in 21 μL of a reaction buffer containing 10 μg of bovine ubiquitin (Sigma-Aldrich), 2 mM ATP, 1 mM MgCl_2_, 0.3 mM dithiothreitol, 25 mM Tris-HCl (pH 7.5), 120 mM NaCl, 5 μM ZnCl_2_, 0–4 μM hemin, 0.5 U of creatine phosphokinase (Sigma-Aldrich), and 1 mM phosphocreatine (Sigma-Aldrich). Reactions were performed for 90 min at 37 °C and terminated by boiling for 5 min in SDS sample buffer containing 5% 2-mercaptoethanol, and the mixtures were then subjected to immunoblot analysis using an anti-HA antibody.

### 4.13. Mass Spectrometry for Phosphorylation Sites Identification

HEK293T cells were transfected with the indicated expression plasmids including FLAG-BACH1, TBK1 and TBK1 S172A. Twenty-four hours after transfection, cells were lysed with a buffer composed of 50 mM HEPES, pH 8, 250 mM KCl, 0.2 mM EDTA, 0.1% Triton X-100, and protease inhibitors. After sonication, lysates were centrifuged and the supernatant was used as whole cell lysates. The suspension was incubated on a rotating platform for 2 h at 4 °C using anti-FLAG beads (M185-11R, MBL, Tokyo, Japan).

Protein identification using the MASCOT search engine was performed as described previously [55]. LC-MS/MS was performed using an Orbitrap Fusion mass spectrometer equipped with an Easy-nLC 1000 HPLC system (Thermo Fisher Scientific, Waltham, MA, USA). The peptides were separated on a C18 tip column (75 μm ID × 10 cm L, Nikkyo Technos, Tokyo, Japan) at a 300 nl/min flow rate with a linear gradient generated by aqueous solvent A (0.1% formic acid in water) and organic solvent B (0.1% formic acid in acetonitrile): 5% B to 35% B in 23.5 min, to 45% B in 27 min, to 95% B in 28 min, 95% B from 28 to 29 min, and finally to 5% B in 30 min. MS1 scans from *m*/*z* = 321 to 1500 were performed in the Orbitrap mass spectrometer with the resolution set to 120000 with lockmasses at *m*/*z* = 445.12003 and 391.28429, which were followed by the acquisition of higher energy collisional dissociation (HCD)-MS2 in the ion trap. The settings for the MS2 scans were as follows: intensity threshold = 1000, charge states = +2 to +6, isolation width = 1.2 *m*/*z*, AGC target = 5000, maximum ion injection time = 50 msec, normalized collision energy = 35% and dynamic exclusion enabled with a 30 s exclusion duration. The MS/MS cycle time was set to 4 s. The MS/MS data were acquired over 35 min after the LC gradient was started. The raw data files were converted to Mascot generic format (mgf) files by Proteome discoverer (version 1.3, Thermo Fisher Scientific) and were submitted to peptide identification using a MASCOT search engine (version 2.5.1, Matrix Science, London, UK) by searching 20693 sequences in total contained in the human proteins in the Swissprot database (November 2020) and a homemade protein list including common contaminating proteins and the mouse Bach1 protein. Trypsin was chosen as the protease, allowing a maximum of three miscleavages. The peptide mass tolerance and MS/MS tolerance were set at 5 ppm and 0.5 Da, respectively. Propionamidated cysteine (+71.0371) was set as a fixed modification, and protein N-terminal acetylation (+42.0106 Da), oxidation of methionine (+15.9949 Da), and phosphorylation at Ser/Thr (+79.9663 Da) were chosen as variable modifications. The threshold of Mascot expectation value for significant peptide-spectral matches was set to 0.05. The false discovery rates estimated by Mascot decoy search were reported as follows: 45 PSM in decoy database vs. 871 in real database (5.16%) for BACH1, 39 vs. 770 (5.06%) in BACH1/TBK1-S172A, and 47 vs. 883 (5.32%) for BACH1/TBK1-wt. The result from a single experiment is shown. 

### 4.14. Plasmids

The expression plasmids were constructed as shown in Appendix A.

### 4.15. Statistics

Statistical analyses were performed using GraphPad Prism 8. For all experiments, differences in data sets were considered statistically significant when *p*-values were lower than 0.05. When comparing only two groups, an unpaired Student’s *t*-test was performed. To compare multiple groups, one-way ANOVA was used. *, *p* < 0.05; **, *p* < 0.01, ns (not significant) *p* > 0.05.

## 5. Conclusions

Our present results show that TBK1 regulates the turnover of BACH1 protein via protein kinase-dependent and -independent mechanisms. It will be an important issue to understand how the balance of the two mechanisms of TBK1-mediated regulation of BACH1 is coordinated or switched in the context of iron metabolism, hematopoiesis and immune response, and cancer progression.

## Figures and Tables

**Figure 1 ijms-25-04141-f001:**
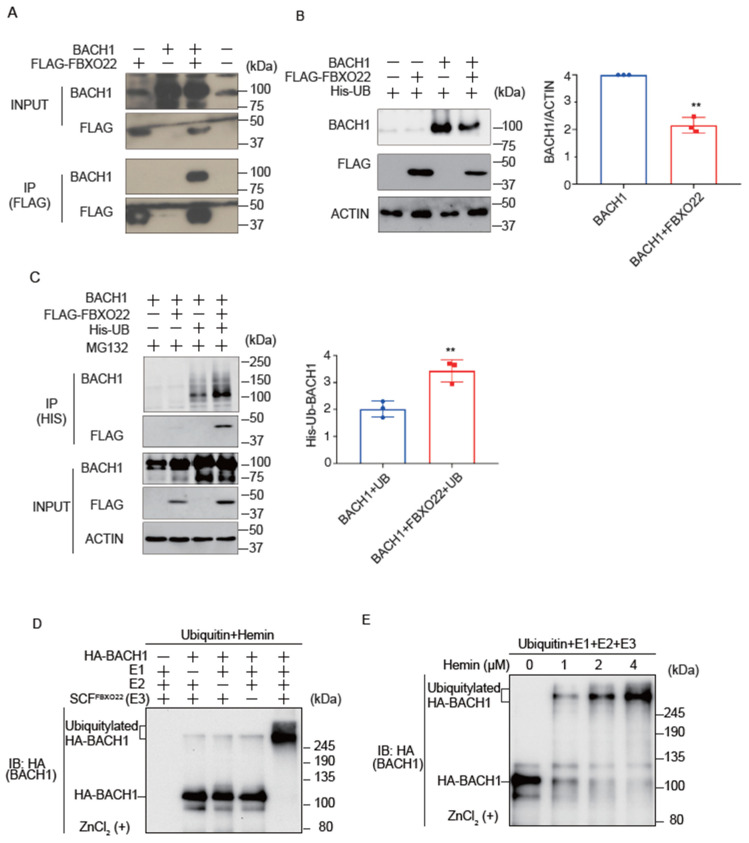
FBXO22 promotes BACH1 degradation and ubiquitination in response to heme. (**A**) Co-immunoprecipitation using an anti-FLAG antibody under a native condition. HEK293T cells were transfected with the indicated expression plasmids. Twenty-four hours after transfection, the whole cell lysates (INPUT) and immunoprecipitated proteins (IP) were examined by protein immunoblotting. (**B**) HEK293T cells were transfected with the indicated expression plasmids for 48 h (**left**). Quantification of BACH1 protein levels is shown as mean ± SD, with *p* values from the Student’s *t*-test (**right**). n = 3 biological replicates. **, *p* < 0.01. (**C**) HEK293T cells were transfected with the indicated expression plasmids and then incubated with MG132 (10 μM) for 12 h. Twenty-four hours after transfection, His-tagged ubiquitin was pulled down with nickel-charged magnetic agarose beads. The original cell lysates (INPUT) and immunoprecipitated proteins (IP) were examined by protein immunoblotting with indicated antibodies (**left**). Quantification of ubiquitinated BACH1 protein levels is shown (**right**) as mean ± SD, with *p* values from the Student’s *t*-test. n = 3 biological replicates. **, *p* < 0.01. (**D**) In vitro ubiquitylation reactions were performed for 90 min with ATP, ubiquitin, hemin (4 μM) and the indicated combinations of HA-BACH1, E1, E2 (CDC34 and UBCH5C), and E3 (SCF-FBXO22), after which reaction mixtures were subjected to immunoblot analysis with antibody to BACH1. (**E**) In vitro ubiquitylation reactions were performed as above with varying concentrations of hemin (0~4 μM).

**Figure 2 ijms-25-04141-f002:**
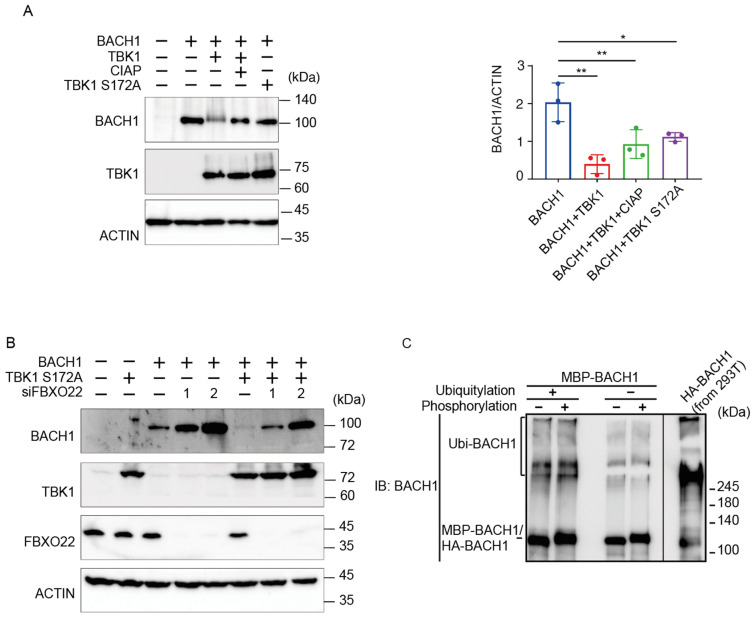
TBK1 promotes BACH1 degradation without FBXO22. (**A**) HEK293T cells were transfected with the indicated expression plasmids. Cell lysates were prepared 48 h after transfection and then treated with CIAP for 1 h where indicated. Western blottings were carried out with indicated antibodies (**left**). Quantification of BACH1 protein levels is presented as mean ± SD, with *p* values obtained from the one-way ANOVA (**right**). *, *p* < 0.05; **, *p* < 0.01. (**B**) Twenty-four hours after TBK1 knockdown, HEK293T cells were transfected with the indicated expression plasmids and cultured for another twenty-four hours. Immunoblottings were carried out using the indicated antibodies. (**C**) MBP-BACH1 was phosphorylated with TBK1 and used as a substrate of in vitro ubiquitylation reactions as above. Ubiquitylated HA-BACH1 was shown as a control.

**Figure 3 ijms-25-04141-f003:**
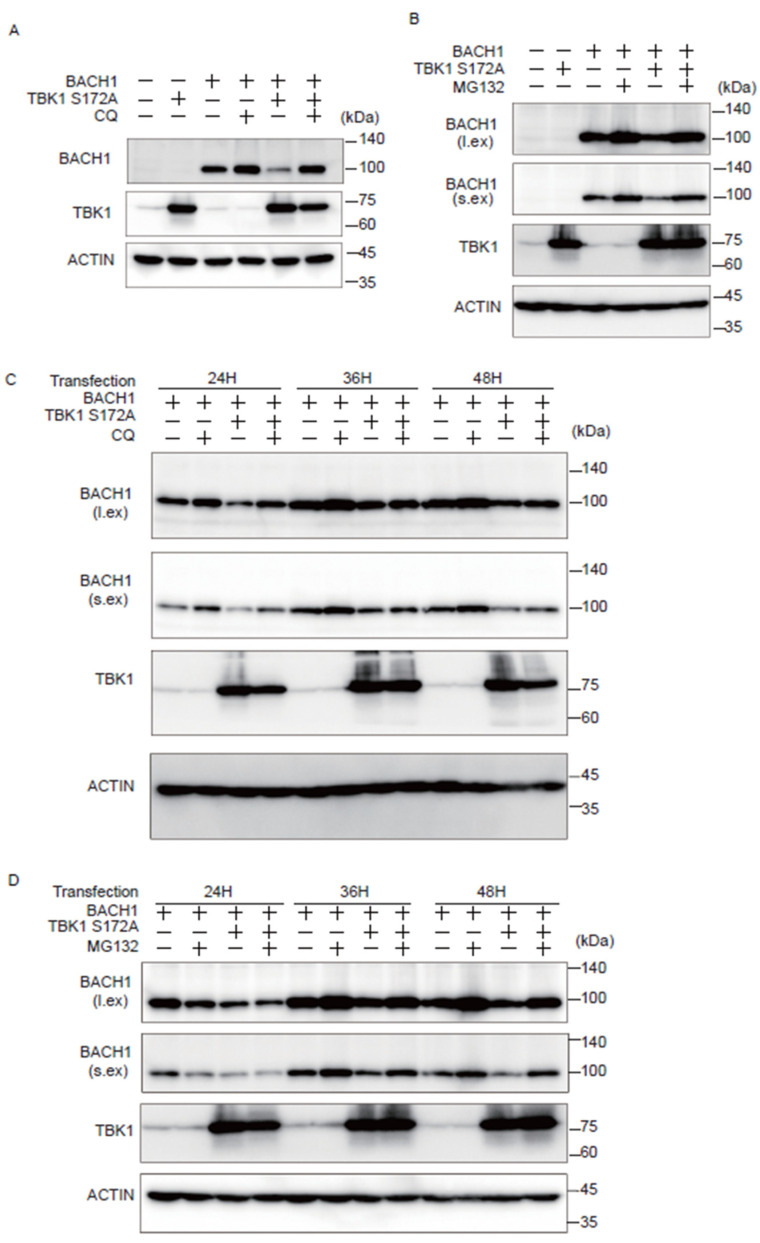
TBK1 S172A promotes BACH1 degradation through autophagy-lysosome and ubiquitin-proteasome pathways. (**A**,**B**) HEK293T cells were transfected with the indicated expression plasmids and then were incubated with CQ (100 μM) or MG132 (10 μM) for 12 h. Cell lysates were prepared at 24 h (**A**) and 48 h (**B**) after transfection for Western blotting with indicated antibodies. l.ex., long exposure; s.ex., short exposure. (**A**–**D**) HEK293T cells were transfected as written above and then were incubated for the indicated duration. Before harvest, the cells were incubated with CQ (100 μM) or MG132 (10 μM) for 12 h. l.ex., long exposure; s.ex., short exposure.

**Figure 4 ijms-25-04141-f004:**
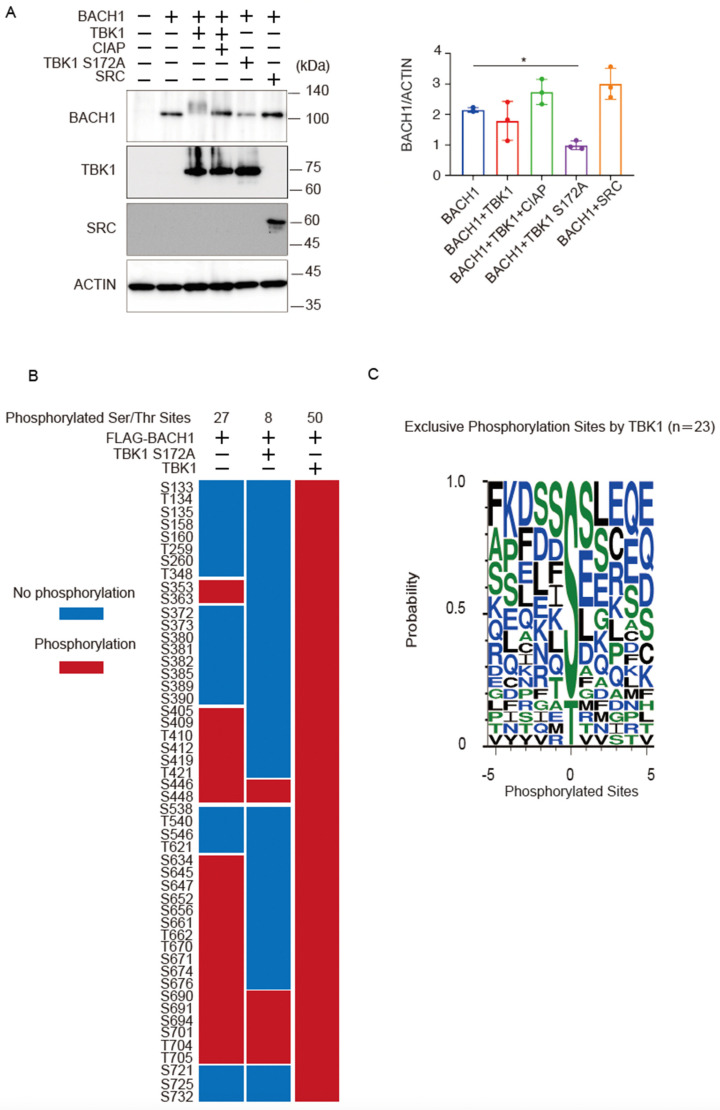
TBK1 phosphorylates BACH1. (**A**) HEK293T cells were transfected with the indicated expression plasmids. Twenty-four hours after transfection, cell lysates were prepared and then treated with CIAP for 1 h where indicated. Western blottings were carried out with indicated antibodies (**left**). Quantification of BACH1 protein levels is presented as mean ± SD, with *p* values obtained from the one-way ANOVA (**right**). *, *p* < 0.05. (**B**) The phosphorylated Ser/Thr residues of mouse BACH1 were identified by MS. HEK293T cells were transfected with the indicated expression plasmids. Twenty-four hours after transfection, FLAG-BACH1 protein was immunoprecipitated using anti-FLAG beads. Shown are the phosphorylated sites assigned with MASCOT expectation values < 0.05. The phosphorylated and unphosphorylated amino acids are color-coded in red and blue, respectively. Considering the diverse phosphorylation states of a peptide containing multiple Ser/Thr residues, the phosphorylation sites assigned a peptide rank > 1 were also included. Therefore, it should be noted that multiple phosphorylation sites were often indicated as a result of a single MS/MS. S, Ser; T, Thr. (**C**) Weblogo of FLAG-BACH1 phosphorylation sites. A weblogo was created by submitting the phosphorylation sites of FLAG-BACH1 by TBK1 to the website http://weblogo.threeplusone.com/create.cgi, accessed on 1 March 2022. The 23 sites that appeared only in the wild-type TBK1-expression sample were considered as the possible TBK1 target sites.

**Figure 5 ijms-25-04141-f005:**
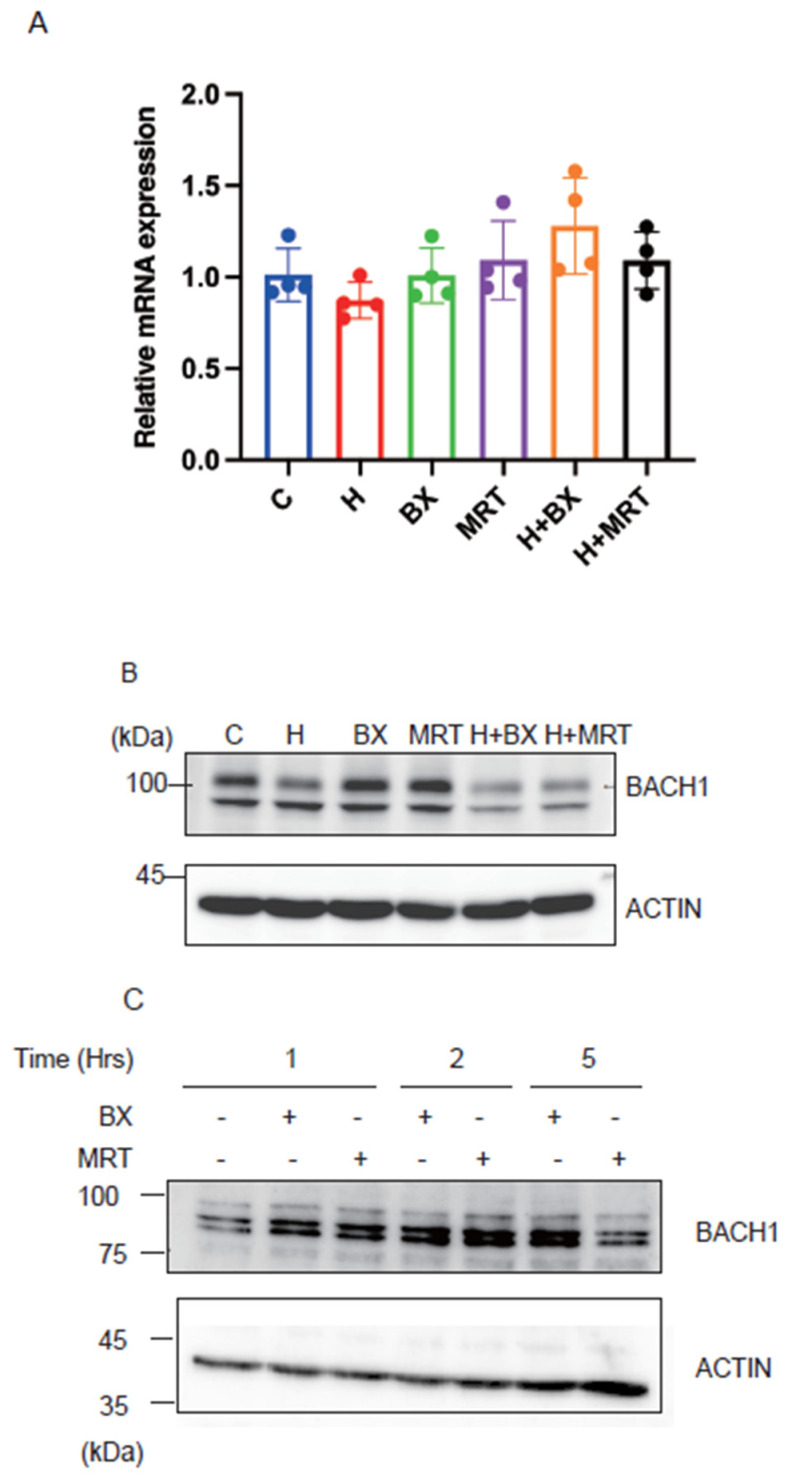
TBK1 maintains the amount of BACH1 protein. (**A**) Relative mRNA levels of *Bach1* in Hepa1 cells which were treated with TBK1 inhibitors (BX795 or MRT67307) and/or hemin (H) for 6 h and control Hepa1 cells. All data are presented as mean ± SD, *n* = 4 biological replicates for each experiment. (**B**) Hepa1 cells were treated with TBK1 inhibitors and/or hemin in indicated combinations for 6 h. Western blots with the antibodies are shown. (**C**) Namalwa cells were treated with TBK1 inhibitors for the indicated periods. Western blots were as in (**B**).

## Data Availability

The raw data files of mass spectrometry are available in the MassIVE public database (https://massive.ucsd.edu/ProteoSAFe/static/massive.jsp, Dataset ID: MSV000090316, accessed on 1 February 2024.). The raw data files of RNA-seq for TBK1 siRNA in AsPC-1 cells are registered with Gene Expression Omnibus (GEO, https://www.ncbi.nlm.nih.gov/geo/, accessed on 1 February 2023) in our previous study [37]. The registration number is GSE201307.

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
