# Peer review of "TANK Binding Kinase 1 Promotes BACH1 Degradation through Both Phosphorylation-Dependent and -Independent Mechanisms without Relying on Heme and FBXO22"

_ijms, 2024, doi:10.3390/ijms25084141_

Round 1

Reviewer 1 Report

Comments and Suggestions for Authors

This manuscript describes the authors' investigation into the degradation of BACH1 through multiple different pathways. Specifically, the authors were interested in detailing a heme-independent protein degradation pathway for BACH1 since some BACH1 target genes are altered in responses to alternative signals. Through a series of experiments, the authors demonstrate that TANK binding kinase 1 (TBK1) degrades BACH1 through a heme-independent pathway. However, the degradation pathway is not elucidated. Thus, the manuscript feels incremental and seems to only add confusion as to how BACH1 is targeted for degradation.

1. Most importantly, the figures need to be updated. The legends do not provide enough information to understand the figures, there is information in the figures that is not explained in the legend or the text (like the unfortunately labeled "s.ex"), the pairwise comparisons are not drawn so that the reader can understand what data the P values are comparing, and quantification of the western blots is consistently provided so the reader can doesn't have to guess at what is significantly different (Fig3C, Fig 3D, Fig4B, and Fig4C in particular).

2. In some figures, but not all, there are two bands for BACH1 in the western blots. Is one of the bands nonspecific? If so, it would be good to point that out to the reader. But then why is not in all of the westerns?

3. The addition of the FBXO22 information in figure 1 feels extraneous and adds to the feeling that the manuscript is unfocused.  Could a model be built that includes FBXO22, HOIL-1, and TBK1 to help tie all of the information together? 

4. The conclusion from figure 1 is that polyubiquitination of BACH1 occurs when "BACH1 which is bound by heme." However, the authors did not specifically demonstrate that BACH1 is binding in this assay. Instead, heme was simply added to the assay. Unless the authors can show that variant of BACH1 that cannot bind heme is not polyubiquitinated (which may be difficult given the complex nature of heme binding to BACH1), the authors can only conclude that heme enhances polyubiquitination. 

5. The time dependence of the effect of CQ and MG132 is interesting; has this been observed before? What significance does this have for the role of BACH1 in the cell?

6. Figure 2B indicates that the knockdown of FBXO22, even in the absence of added heme, increased BACH1 levels. It seems then that the data in figures 3-5 could be influenced by the activity of FBXO22, preventing the effect of TBK1 alone from being fully uncovered. For instance, if FBXO22 is knocked down in the CQ/MG132 experiment with wild-type TBK1, would the  effect of these inhibitors then become apparent? Or even growth in heme-depleted media?

Author Response

Please refer to a word file.

Reviewer 2 Report

Comments and Suggestions for Authors

The manuscript “TANK binding kinase 1 promotes BACH1 degradation through 1 both phosphorylation-dependent and -independent mecha-2 nisms without relying on heme and FBXO22” reports the phosphorylation-dependent and -independent degradation of BACH1 by TBK1. The authors performed experiments by using wild-type TBK1 and S172A mutant with reduced phosphorylation activity to investigate their mechanism.

The paper is of interest, but I would like to confirm something about the discussion of the study. My main question is about the interpretation and discussion of the consequences of phosphorylation-dependent BACH1 degradation by the wild-type TBK1 (comments in the following “major points”).

In addition, I felt that the explanation of the results is a little lacking. Since the methods are described in later sections, I think it is better to add a brief explanation of terms in the results as well (comments in the following “minor points”).

So, in my opinion, it would be better that the following points is considered before publication.

* After reading the manuscript, I downloaded it again and found that it had been changed. Therefore, it may contain unnecessary comments.

- Major points

1.    The results in Sections 2.2 and 2.4 show that BACH1 is little phosphorylated and is degraded when cell lysates were prepared 48 hours after transfection, while it is phosphorylated and is little degraded for 24 hours. I may have misunderstood, but from these results, it appears that dephosphorylation after phosphorylation is also important. I think it is better that a discussion of interpretation of the above results and the analysis of phosphorylation site of BACH1 in cells 48 hours after transfection are also added.

2.    In Fig. S2A, it appears that BACH1 phosphorylation is reduced and BACH1 is rather degraded in “24H” and “36H.” These results should also be evaluated quantitatively. Have the authors examined the possibility that CQ and MG132 are involved in the dephosphorylation of BACH1?

3.    On page 13, around line 315, the authors write “when proteasome or lysosome is inhibited, the other phosphorylation-independent mechanism may be induced to compensate for the accumulation of BACH1, directing more BACH1 protein toward proteasome or lysosome.” It is not clear why compensation by the other mechanism direct more BACH1 protein toward proteasome or lysosome and how this leads to a reduction of phosphorylation of BACH1. Is this mean that BACH1 accumulation leads to a reduction of BACH1 phosphorylation? If so, why did the experimental result show a degradation of BACH1 (Fig. S2A)?

4.    The authors write "It will be important issue to understand how the balance of the two mechanisms of TBK1-mediated regulation of…" in Section 4 (conclusion). Both mechanisms promote BACH1 degradation. I think the authors should discuss what specific regulation are possible (timing of degradation?).
Even if this section (conclusion) has been deleted, I think that the difference between BACH1 degradation by TBK1 in the inactivated and activated states should be discussed.

- Minor points

1.    The last author's name may be incorrect.

2.    In Section 2.1, it would be better to have a brief description of the experiment and results of ubiquitylation assay as well as the figure and its caption.

3.    I think it is better to add explanations of terms; "HA-BACH1," (HA-tagged?) "SCF-FBXO22," and "MBP-BACH1." (Ignore this comment if it is unnecessary in this field.)

I couldn't understand the following description in the section 5.12 (materials and methods); "SCF-FBXO22 was prepared from HEK293T cells as HA-BACH1."

4.    The labels overlap in Figs. 2B and C.

5.    In Fig. 4A, the double asterisk (**) described in the caption is missing.

6.    For Fig. 4C, it should be added a horizontal axis label.

7.    I think it would be better to unify the notation of Hep1, Hepa 1, and Hepa1.

8.    In Section 2.6, why is the degradation of BACH1 small for Hemin alone? This result should also be evaluated quantitatively.

9.    In Fig. 5C, why is the location of BACH1 (around 80-90 kDa) different from other experiments (~100 kDa)?

10. The caption for Fig. 6 is probably incorrect.

Comments on the Quality of English Language

English is generally fine. Please check the commented items.

Author Response

Please refer to a word file.

Reviewer 3 Report

Comments and Suggestions for Authors

Liu et al. found that TBK1 could promote BACH1 degradation using a series of assays. The manuscript is well-written. I suggest that the manuscript is accepted for publication after the following revisions:

1. In line 97, the author mentioned that “it remains unclear how heme increases the activity of FBXO22”, but the figure 1 cannot prove this question. And in lines 104 and 105, “in these cells”, which cell? In Figure S1B, too few timepoints are set, five timepoints are necessary, and the quantitative analysis should be also provided.

2. In lines 207 and 208, as protein kinase SRC did not cause a shift 208 or a reduction of the BACH1 band, the effects of TBK1 and TBK1 S172A on BACH1 were specific. This is not accurate. It should do this assay using more unrelated protein kinase.

3. I suggest that this author provides a model diagram of the mechanism.

Author Response

Please refer to a word file.

Round 2

Reviewer 1 Report

Comments and Suggestions for Authors

The authors have responded to the questions and concerns raised in the initial review and improved the manuscript accordingly. 

Reviewer 2 Report

Comments and Suggestions for Authors The authors have addressed almost all my concerns. The overlap between Figures 2B and C (#4 of minor comments) does not appear to have been resolved, but it is trivial and I agree the publication of this manuscript.